# Synthesis of Chitosan-La_2_O_3_ Nanocomposite and Its Utility as a Powerful Catalyst in the Synthesis of Pyridines and Pyrazoles

**DOI:** 10.3390/molecules26123689

**Published:** 2021-06-17

**Authors:** Khaled D. Khalil, Sayed M. Riyadh, Mariusz Jaremko, Thoraya A. Farghaly, Mohamed Hagar

**Affiliations:** 1Department of Chemistry, Faculty of Science, Cairo University, Giza 12613, Egypt; riyadh1993@hotmail.com (S.M.R.); thoraya-f@hotmail.com (T.A.F.); 2Department of Chemistry, Faculty of Science, Taibah University, Al-Madinah Almunawarah, Yanbu 46423, Saudi Arabia; mhagar@taibahu.edu.sa; 3Department of Chemistry, College of Science, Taibah University, Al-Madinah Almunawrah 30002, Saudi Arabia; 4Biological and Environmental Sciences & Engineering Division (BESE), King Abdullah University of Science and Technology (KAUST), Thuwal 23955-6900, Saudi Arabia; Mariusz.jaremko@kaust.edu.sa; 5Department of Chemistry, Faculty of Applied Science, Umm Al-Qura University, Makkah Almukaramah 21514, Saudi Arabia; 6Chemistry Department, Faculty of Science, Alexandria University, Alexandria 21321, Egypt

**Keywords:** chitosan, La_2_O_3_, nanocomposite film, heterogeneous catalysis, pyridines, pyrazoles

## Abstract

Recently, the development of nanocatalysts based on naturally occurring polysaccharides has received a lot of attention. Chitosan (CS), as a biodegradable and biocompatible polysaccharide, is considered to be an excellent template for the design of a hybrid biopolymer-based metal oxide nanocomposite. In this case, lanthanum oxide nanoparticles doped with chitosan at different weight percentages (5, 10, 15, and 20 wt% CS/La_2_O_3_) were prepared via a simple solution casting method. The prepared CS/La_2_O_3_ nanocomposite solutions were cast in a Petri dish in order to produce the developed catalyst, which was shaped as a thin film. The structural features of the hybrid nanocomposite film were studied by FTIR, SEM, and XRD analytical tools. FTIR spectra confirmed the presence of the major characteristic peaks of chitosan, which were modified by interaction with La_2_O_3_ nanoparticles. Additionally, SEM graphs showed dramatic morphological changes on the surface of chitosan, which is attributed to surface adsorption with La_2_O_3_ molecules. The prepared CS/La_2_O_3_ nanocomposite film (15% by weight) was investigated as an effective, recyclable, and heterogeneous base catalyst in the synthesis of pyridines and pyrazoles. The nanocomposite used was sufficiently stable and was collected and reused more than three times without loss of catalytic activity.

## 1. Introduction

With the growing concern of nanocatalysis in organic transformations, naturally occurring biopolymers have been used extensively as powerful substrates to be utilized as an excellent stabilizer for the immobilizing of a variety of metal oxides nanoparticles due to their biodegradable, nontoxic, low cost, and excellent structural properties [1]. Nanocomposite hybrid materials are where various materials combine to develop unique properties guaranteeing that one of the materials has a size in the range of 1–100 nm [2]. Therefore, much effort has been devoted to the design and synthesis of these hybrid nanocomposites through flexible and efficient routes; there has also been careful study of their new properties to adapt them to numerous applications in the areas of materials science, especially their catalytic potency in the organic reactions [3,4,5,6,7].

Chitosan (CS), the partially deacetylated form of chitin, is prepared via alkaline hydrolysis under certain conditions. Chitosan and its derivatives, rather than any other polysaccharide, are considered an excellent template to immobilize metal oxide nanoparticles owing to its unique structural features, in particular the presence of the hydroxyl and amino groups [8,9].

On the other hand, lanthanum oxide (La_2_O_3_) nanoparticles have shown considerable basic properties that adapt to many base catalyzed reactions [10,11]. Unfortunately, some difficulties limit its use, such as its difficult separation and reusability since the used catalyst could not be quantitatively recovered and the purification of the products being the biggest challenge.

Extending our earlier efforts in the area of nanocatalysis [5,6,7], in this article La_2_O_3_ nanoparticles immobilized onto the chitosan matrix were prepared (Figure 1) and then employed as a promising heterogeneous basic catalyst to synthesis pyridines and pyrazoles. These azines and azoles have privileged antitumor [12,13,14,15], antidiabetes [16], antimicrobial [17,18], anti-HCV [14,19,20], and antidepressant [21,22] activities.

## 2. Results and Discussion

### 2.1. Preparation and Characterization of CS/La_2_O_3_ Nanocomposite Film

Chitosan-La_2_O_3_ (CS/La_2_O_3_) nanocomposite films were prepared using a coprecipitation method using chitosan as a biostabiliser [5,6]. The chitosan solution in acetic acid was treated with the appropriate amount of La_2_O_3_ nanopowder, and then the solvent was evaporated at ambient temperature. The prepared nanocomposite films were studied by FTIR, SEM, and XRD as follows:

#### 2.1.1. FTIR Characterization

The FTIR spectra of the chitosan (A), La_2_O_3_ nanoparticles (B), and chitosan-La_2_O_3_ nanocomposite (C) were measured and depicted in Figure 2. Figure 2A exhibited the presence of main characteristic bands of chitosan, namely, at υ = 3368 cm^−1^ (OH- group), 2912, 2874 cm^−1^ (C-H bond; CH_3_ groups), 1602 cm^−1^ (amide carbonyl groups), 1381 cm^−1^ (CH_2_ groups), and 1057 cm^−1^ (C-O) [5,6]. In addition, Figure 2B shows the main characteristic bands of the La_2_O_3_ at 1523, 1362, and 829 cm^−1^ [23]. For comparison, Figure 2C shows mixed bands as result of the combination of the chitosan structure with the La_2_O_3_ nanoparticles, which is attributed to the presence of a chemical interaction between La_2_O_3_ molecules and the binding sites of OH and NH_2_ groups of chitosan.

#### 2.1.2. SEM and Morphological Changes

A scanning electron microscope (SEM) was used to confirm the presence of morphological changes in the surface caused by the incorporation of La_2_O_3_ into the chitosan matrix. The micrographs of the pure chitosan (A), La_2_O_3_ nanoparticles (B), and the hybrid nanocomposite films (C) with 15 wt% are shown in Figure 3. As can be seen, there is a marked morphological change in the fibrous surface of chitosan, as opposed to the chitosan-La_2_O_3_ hybrid nanocomposite films. According to the SEM images, La_2_O_3_ nanoparticles were homogenously distributed over the whole surface of the chitosan matrix, and the average size of La_2_O_3_ particles was found to be approximately 30 nm for 15 wt%. Moreover, an energy dispersive X-ray (EDX) of the CS/La_2_O_3_ nanocomposite film showed the presence of La_2_O_3_ inside the polymer matrix (Figure 4). Thus, from the following EDX, elemental analysis of the nanocomposite determined the La_2_O_3_ content to be ~15 wt%.

#### 2.1.3. X-ray Diffraction

The X-ray diffraction (XRD) measurement was performed to study the crystallinity and nanostructure features of the native chitosan and the chitosan-La_2_O_3_ (15 wt%) thin film nanocomposites. In Figure 5, the XRD of blank chitosan (A) shows the main characteristic broad peak at 2θ = 20°, which indicates the presence of crystalline and amorphous regions in chitosan as reported in the literature [12]. Moreover, the purity of chitosan was confirmed by the absence of any additional peaks of impurities in the pattern. On the other hand, the XRD of La_2_O_3_ nanoparticles (B) exhibits normal peaks at 26°, 28.5°, 30.2°, and 39.4°, which is in agreement with the results reported in [24]. For the doped nanocomposite film (15 wt% CS/La_2_O_3_) (C), a combination of the same peaks of the individual components appeared, but in different intensities, indicating there was an interaction between La_2_O_3_ molecules and the OH and NH_2_ groups of the chitosan chain. The average grain size was calculated from the XRD patterns using the Debye–Scherrer formula [25].
$$D\left(nm\right)=\frac{-0.9\times \lambda }{\beta \times \cos\theta }$$
where D(nm) is the crystalline size in nm, λ is the wavelength of Cu-kα1 = 1.54060 A°, and β can be calculated for the most intense peak for the CS/La_2_O_3_ nanocomposite pattern. The average particle size was found to be 32.2 nm.

### 2.2. CS/La_2_O_3_ Nanocomposite Film as Basic Catalyst in Synthesis of Pyridine Derivatives

The reaction between malononitrile dimer **1** with enaminone **2a** in absolute ethanol was conducted in presence of the CS/La_2_O_3_ nanocomposite in order to estimate the optimized conditions and the proper catalyst loading (Scheme 1).

#### Optimizing the Catalyst Loading and the Reaction Conditions

In order to estimate the proper catalyst loading, a model reaction of malononitrile dimer **1** with enaminone **2a** was carried out using 5, 15, 20, and 25 wt% of nanocomposite film under the same conditions. From the obtained results, the catalyst loading 15 wt% was found to be the optimal quantity for the maximum progress of the reaction (90% yield) after refluxing for 2 h (Figure 6). Moreover, the recovered catalyst was successfully used three times without significant change in its catalytic potency (Table 1).

Pyridine derivatives **3a**–**h** with different aromatic moieties were prepared successfully via mixing of malononitrile dimer **1** with enaminones **2a**–**h** in absolute ethanol and in the presence of basic catalyst (piperidine or chitosan, La_2_O_3_ or CS/La_2_O_3_ nanocomposite), in a comparable yield (Scheme 2, Table 2). The results show that the use of CS/La_2_O_3_ nanocomposite afforded good yields of the products when utilized as a basic promoter over piperidine, chitosan, or La_2_O_3_ under similar employed conditions. The use of La_2_O_3_ nanoparticles has obvious technical problems, such as the difficulty of its recovery from the reaction medium; the contamination of the product is also a significant problem that may affect the accuracy of the % yield calculation. Moreover, the chitosan-based nanocatalyst was found to be superior in comparison to the others, not only due to its synergistic action but also owing to the cross-linking nature of the hybrid nanocomposite that facilitated its removal from the reaction medium by simple filtration and, consequently, its accessibility to reuse after washing with water and ethanol several times without loss of its catalytic activity.

In the following reasonable mechanism for the above-mentioned reaction (Scheme 3), La_2_O_3_ nanoparticles acted as a Bronsted base [10,11] which deprotonated the active methylene group of malononitrile dimer (**1**) and gave the respective anion intermediate **A**. The latter carbanion attacked the β-carbon of enaminone (**2a**–**h**) with displacement of dimethylamine, as a good leaving group, to give the non-isolable intermediate **B**. Intramolecular condensation of intermediate **B** afforded the pyridine derivatives **3a**–**h**.

The green protocol for the efficient synthesis of functionalized azoles was extended. Thus, the reaction of 2-arylhydrazone-malono-1,3-dial (**4a**–**c**) with chloroacetone (**5**) under refluxing conditions using different base catalysts (piperidine, chitosan, La_2_O_3_, or CS/La_2_O_3_ nanocomposite) proceeded smoothly to provide the corresponding pyrazoles **6a**–**c** in a comparable yield (Scheme 4, Table 3). Again, the results show that CS/La_2_O_3_ is superior in acting efficiently as a basic promoter for the reaction due to its synergistic effect as well as its ease of recovery and recycling from the reaction medium.

From the results, the toxic triethyl amine can be replaced successfully by the greener nanocomposite that gave the product in a relatively similar yield.

The latter reaction was promoted by the presence of the basic catalyst CS/La_2_O_3_, which abstracted proton from arylazohydrazonals (**4a**–**c**) to initiate the nucleophilic displacement of chlorine atoms and provided intermediate A. Further dehydrogenation followed by base catalyzed cyclization gave intermediates B and C. Dehydration of intermediate C provided the isolable pyrazoles **6a**–**c** (Scheme 5).

## 3. Experimental

### 3.1. Materials and Methods

Sigma-Aldrich provided the chitosan (powder, shrimp shells, product no. C3646, density = 0.15–0.3 g/cm^3^) and La_2_O_3_ (nano powder, TEM particle size 100 nm, product no. 634271). The FTIR (Fourier-Transform Infrared) spectra of nanocomposite were captured using potassium bromide discs by a Pye-Unicam SP300 Instrument (Cambridge, UK). On a high-resolution scanning electron microscope (model HRSEM, JSM 6510A, Jeol, Tokyo, Japan), SEM analyses were recorded. A Philips Diffractometer (Model: X’Pert-Pro MPD; Philips, Eindhoven, The Netherlands) was used to calculate XRD. On an electrothermal Gallenkamp capillary apparatus (Leicester, UK), the melting points of samples were determined. On a Varian Mercury VXR-300 spectrometer (300 MHz for ^1^H NMR and 75 MHz for ^13^C NMR), the chemical shifts of the newly synthesized compounds in DMSO-*d*_6_ were recorded. The mass spectra were recorded on a GCMSQ1000-EX Shimadzu and GCMS 5988-A HP spectrometers; the ionizing voltage was 70 eV.

### 3.2. Preparation of CS/La_2_O_3_ Nanocomposite Film

A 2 wt% solution of medium molecular weight chitosan was prepared by dissolving it in a 2% (*w*/*v*) aqueous acetic acid solution and stirring for 48 h at room temperature. The resulting viscous solution was filtered to obtain the homogeneous clear chitosan solution. A quantity of this solution was taken in a 50 mL bottle and 5, 10 and 15, 20 (*w*/*v*%) of La_2_O_3_ was added portion-wise with vigorous stirring, and then the stirring was continued for an additional 24 h. To remove the solvent, the solution was poured into a Teflon Petri dish (8 cm) and dried in a vacuum oven set to 50 °C for 3 days. After neutralization with 5 mL of 1 M NaOH, the chitosan-La_2_O_3_ nanocomposite film was peeled off the Petri dish and rinsed with distilled water. Finally, the film was held for two days at room temperature in a vacuum desiccator.

### 3.3. Reaction of Malononitrile Dimer with Enaminone

Method A: Both 2-aminoprop-1-ene-1,3,3-tricarbonitrile (**1**) (1.32 g, 0.01 mol) and enaminones **2a**–**h** (0.01 mol) were mixed in 10 mL of absolute ethanol and subsequently treated with an appropriate amount (5 drops) of the piperidine as a base catalyst. The reaction mixture was refluxed until all the starting materials were completely consumed (within 2 h as monitored by TLC). When the reaction was finished, the mixture was cooled and poured over the water–ice mixture while stirring. The precipitate was filtered, washed with water and crystallized from ethanol as yellow crystals.

Method B: The same procedure as that of method A was applied under the same conditions, but using CS, La_2_O_3_, or CS/La_2_O_3_ nanocomposite film (15 wt%) instead of the piperidine. Once the reaction was complete, the film was carefully removed and washed with water and ethanol for multiple uses in other reactions.

*2-(3-Cyano-6-phenyl-1H-pyridin-2-ylidene)malononitrile* (**3a**) [26]. m.p. 257−259 °C; Anal. Calcd. C_15_H_8_N_4_ (244.26): C, 73.76; H, 3.30; N, 22.94. Found: C, 73.68, H, 3.24; N, 22.87%. Accurate mass: 244.146.

*2-(3-Cyano-6-(4-methylphenyl)-1H-pyridin-2-ylidene)malononitrile* (**3b**) [27]. m.p. 297–298 °C; Anal. Calcd. C_16_H_10_N_4_: C, 74.40; H, 3.90; N, 21.69. Found: C, 74.31; H, 3.76; N, 21.62%. Accurate mass: 258.172.

*2-(3-Cyano-6-(4-methoxyphenyl)-1H-pyridin-2-ylidene)malononitrile* (**3c**) [28]. m.p. 275−277 °C; Anal. Calcd. C_16_H_10_N_4_O: C, 70.06; H, 3.68; N, 20.43. Found: C, 69.93; H, 3.61; N, 20.37%. Accurate mass: 274.015.

*2-(3-Cyano-6-(4-chlorophenyl)-1H-pyridin-2-ylidene)malononitrile* (**3d**) [28]. m.p. 255–256 °C; Anal. Calcd. C_15_H_7_ClN_4_: C, 64.65; H, 2.53; Cl, 12.72; N, 20.10. Found: C, 64.48; H, 2.49; Cl, 12.67; N, 20.03%. Accurate mass: 278.719.

*2-(3-Cyano-6-(4-nitrophenyl)-1H-pyridin-2-ylidene)malononitrile* (**3e**). m.p. 194−196 °C; IR (KBr, cm^−1^): 3250 (NH), 2163 (3 CN); ^1^H-NMR (DMSO-*d*_6_): *δ*, ppm = 7.11–7.82 (m, 6H, Ar-H), 9.47 (br, 1H, NH, D_2_O exchangeable); ^13^C-NMR (DMSO-*d*_6_): *δ*, ppm = 184.3, 164.2, 161.7, 155.9, 130.1 (2C), 128.9, 128.5, 117.3, 114.9, 110.2, 83.3, 62.7. Anal. Calcd. C_15_H_7_N_5_O_2_: C, 62.29; H, 2.44; N, 24.21. Found: C, 62.17; H, 2.35; N, 23.89%. Accurate mass: 289.104.

*2-(3-Cyano-6-furan-2-yl-1H-pyridin-2-ylidene)malononitrile* (**3f**). m.p. 182–183 °C; IR (KBr, cm^−1^): 3263 (NH), 2157 (3 CN); ^1^H-NMR (DMSO-*d*_6_): *δ*, ppm = 7.02–7.69 (m, 5H, Ar-H), 9.68 (br, 1H, NH, D_2_O exchangeable); ^13^C-NMR (DMSO-*d*_6_): *δ*, ppm = 174.1, 160.6, 156.1, 151.0, 141.4, 129.2 (2C), 126.7, 117.4, 116.9 (2C), 106.8, 58.0. Anal. Calcd. C_13_H_6_N_4_O: C, 66.67; H, 2.58; N, 23.92. Found: C, 66.37; H, 2.48; N, 23.74%. Accurate mass: 234.067.

*2-(3-Cyano-6-thiophen-2-yl-1H-pyridin-2-ylidene)malononitrile* (**3g**) [26]. m.p. 200−202 °C; Anal. Calcd. C_13_H_6_N_4_S: C, 62.39; H, 2.42; N, 22.39; S, 12.81. Found: C, 62.32; H, 2.37; N, 22.26; S, 12.72%. Accurate mass: 250.043.

*2-[3-Cyano-6-(1H-pyrrol-2-yl)-1H-pyridin-2-ylidene]-malononitrile* (**3h**). m.p. 167−168 °C; IR (KBr, cm^−1^): 3255 (2 NH), 2168 (3 CN); ^1^H-NMR (DMSO-*d*_6_): *δ*, ppm = 5.84 (br, 1H, pyrrole-NH, D_2_O exchangeable); 6.96–7.52 (m, 5H, Ar-H), 9.10 (br, 1H, pyridine-NH, D_2_O exchangeable); ^13^C-NMR (DMSO-*d*_6_): *δ*, ppm = 168.2, 156.9, 152.6, 148.2, 139.0, 127.8 (2C), 127.4, 117.8, 117.3(2C), 104.1, 51.5. Anal. Calcd. C_13_H_7_N_5_: C, 66.95; H, 3.03; N, 30.03. Found: C, 66.84; H, 2.96; N, 29.91%. Accurate mass: 233.163.

### 3.4. Reaction of 2-arylazomalonaldehyde 6 with Chloroacetone

Method A: A mixture of arylazomalonaldehyde **4a**–**c** (10.0 mmol) and *α*-chloroacetone (**5**) (0.92 g, 10.0 mmol) was dissolved in 20 mL ethanol, containing a few drops of triethylamine. The reaction mixture was stirred for 2 h at room temperature, and the solid material was filtered and washed using a small quantity of ethanol. The crude products **6a**–**c** were purified by recrystallization from ethanol.

Method B: The same procedure as that of method A was applied under the same conditions, but with the aid of CS, La_2_O_3_, or CS/La_2_O_3_ nanocomposite film instead of the triethylamine. After completion of the reaction, the film was carefully removed and was then washed with water and ethanol for several uses in other reactions.

*5-Acetyl-1-phenyl-1H-pyrazole-3-carbaldehyde* (**6a**). m.p. 164–166 °C. IR (KBr): *v* = 1703, 1683 (2 C=O) cm^−1^. ^1^H NMR (DMSO-*d*_6_): *δ*, ppm = 2.72 (s, 3H, COCH_3_); 7.34–7.59 (m, 5H, Ar-H); 7.89 (s, 1H, pyrazole-H); 9.84 (s, 1H, CHO). MS: *m*/*z* (%) = 214.1 (100) [M]^+^. Anal. Calcd. C_12_H_10_N_2_O_2_: C, 67.28; H, 4.71; N, 13.08. Found C, 67.12; H, 4.61; N, 12.93.

*5-Acetyl-1-(4-chlorophenyl)-1H-pyrazole-3-carbaldehyde* (**6b**) [29]. m.p. 180–181 °C; MS: *m*/*z* (%) = 248 (80) [M]^+^; Anal. Calcd. C_12_H_9_ClN_2_O_2_: C, 57.96; H, 3.65; Cl, 14.26; N, 11.27. Found C, 57.84; H, 3.56; Cl, 14.18; N, 11.13.

*5-Acetyl-1-(4-methoxyphenyl)-1H-pyrazole-3-carbaldehyde* (**6c**). m.p. 152–153 ^o^C. IR (KBr): *v* = 1705, 1690 (2 C=O) cm^−1^. ^1^H NMR (DMSO-*d*_6_): *δ*, ppm = 2.58 (s, 3H, COCH_3_); 3.75 (s, 3H, OCH_3_); 7.17–7.25 (d, 2H, *J* = 8.0 Hz, Ar-H), 7.44–7.56 (d, 2H, *J* = 8.0 Hz, Ar-H); 7.88 (s, 1H, pyrazole-H); 10.32 (s, 1H, CHO). MS: *m*/*z* (%) = 244.1 (80) [M]^+^. Anal. Calcd. C_13_H_12_N_2_O_3_: C, 63.93; H, 4.95; N, 11.47. Found C, 63.84; H, 4.86; N, 11.32.

## 4. Conclusions

In this study, FTIR, FESEM, and EDX spectra were used to characterize the preparation of a chitosan-La_2_O_3_ nanocomposite (as a green recyclable biocatalyst). For 15 wt%, the average size of the La_2_O_3_ particles was found to be about 30–32 nm. In a comparison study with respect to triethylamine as a conventional catalyst with chitosan, this hybrid nanocomposite film worked well as a heterogeneous catalyst for the synthesis of pyridines and pyrazoles. In addition to having a better environmental impact, the chitosan-La_2_O_3_ nanocomposite was found to be a more effective catalyst in these reactions than triethylamine. From the catalytic studies, the synergistic effect produced by the combination of the basic nature of both La_2_O_3_ nanoparticles and chitosan itself was the main reason for the superior catalytic potency of the chitosan-La_2_O_3_ nanocomposite as compared to the use of its individual components. In addition to its green impact, the nanocatalyst film could also be easily removed, restored, and reused without losing its catalytic activity. Finally, the chitosan–metal oxide hybrid nanocomposite is a promising hybrid nanocomposite that needs to be investigated further in a variety of organic transformations.

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
