# Peer review of "Synthesis of Chitosan-La2O3 Nanocomposite and Its Utility as a Powerful Catalyst in the Synthesis of Pyridines and Pyrazoles"

_molecules, 2021, doi:10.3390/molecules26123689_

Round 1

Reviewer 1 Report

I believe the paper should be published but the minor revision is needed.

When discussing the results, the following issues should be clarified:

1) Can the molecular weight of chitosan and the degree of its deacetylation affect the stability and activity of the catalyst?

2) Is it possible to estimate the fraction of particles of the active phase La2O3 available for reaction? Was the number of active sites estimated by the chemisorption method? If such studies were carried out, the results should be presented in the article.

3) It is necessary to clarify the sizes of La2O3 nanoparticles.

Page 8, line 174. The particle size of the initial La2O3 is 100 nm.

Page 3, line 90. According to the SEM results, the average particle size of La2O3 is 20 nm.

Page 10, line 265. In conclusion, the size of La2O3 particles is 6-11 nm.

4) Page 3, lines 88-90. It is desirable to confirm the information contained in the text with high-resolution TEM images.

Figure 1 contains no useful information and can be deleted.

Bets regards,

Author Response

Response to Reviewer 1 Comments

Point 1: Can the molecular weight of chitosan and the degree of its deacetylation affect the stability and activity of the catalyst?

Response 1: Actually, from our previous work we have selected the medium molecular weight grade of chitosan due to:

  1. The ease of its dissolution, and consequently the formation of homogeneous solution during the preparation of Nano composite film.
  2. The medium molecular weight chitosan afforded an excellent nanocomposite film that has a relatively higher stability than that of the low molecular weight grade.

Point 2: Is it possible to estimate the fraction of particles of the active phase La2O3 available for reaction? Was the number of active sites estimated by the chemisorption method? If such studies were carried out, the results should be presented in the article.

Response 2: Unfortunately, such study needs extra unavailable tools and our catalytic reactions herein based on the basic nature of both chitosan and La2O3 molecules that synergistically act enough as an efficient base promoter for the reactions under investigation. But we will do in our perspective research points.

Point 3: It is necessary to clarify the sizes of La2O3 nanoparticles.

Page 8, line 174. The particle size of the initial La2O3 is 100 nm.

Page 3, line 90. According to the SEM results, the average particle size of La2O3 is 20 nm.

Page 10, line 265. In conclusion, the size of La2O3 particles is 6-11 nm.

Response 3: Thank you very much for valuable comment.

The starting La2O3 nanoparticles were purchased and the sample delivered was defined as nanoparticles < 100 nm. But during the reaction the size of La2O3 may be varied after dissolution and chemical interaction with the chitosan active sites, that is why the size is changed to less value as compared to that of the original (30 nm (not 20 nm) based on SEM and 32.2 nm Based on the XRD analysis). The size of nanoparticles corrected in the text (SEM and Conclusion).

Point 4: Page 3, lines 88-90. It is desirable to confirm the information contained in the text with high-resolution TEM images.

Response 4: Thank you very much for your accurate comment, we agree to the reviewer comment that the TEM will be more informative. However, the TEM is now unavailable to do. Alternatively, we introduced XRD and the SEM  for the  measurements of particle size based of the prepared catalyst .

Point 5: Figure 1 contains no useful information and can be deleted.

Response 5: Ok, the figure is deleted upon the reviewer request.

Reviewer 2 Report

The authors developed chitosan-La2O3 nanocomposite via a simple solution casting method, which have been applied on the synthesis of pyridines and pyrazoles. The results are interesting and recommend to publish on the Molecules after major revision. The following aspects need to be considered for improvement.

  1. Please compare the catalytic activities of chitosan, La2O3 NPs, and chitosan-La2O3 composite in the synthesis of pyridines and pyrazoles, confirm the function of chitosan and La2O3, respectively.
  2. In the synthesis of pyridines and pyrazoles, the by-products should be referred in the manuscript.
  3. 15%wt should be changed into 15wt%.
  4. The products should be detected by NMR.
  5. Please provide some structure information of the spent chitosan-La2O3 composite catalyst.

Author Response

Response to Reviewer 2 Comments

Point 1: Please compare the catalytic activities of chitosan, La2O3 NPs, and chitosan-La2O3 composite in the synthesis of pyridines and pyrazoles, confirm the function of chitosan and La2O3, respectively.

Response 1: Actually, our research studies based on the use of recyclable catalysts thus the use of La2O3 is excluded for the contamination reasons of the product and the difficult of its isolation and recovery from the reaction medium. Regarding, the role of chitosan and chitosan-La2O3catalyst, the basic character of both substances is the most effective role of them, where the chitosan is basic in nature owing to the presence of amino groups along the chain. The traditional method for such synthesis used triethyl amine and this catalyst was very harmful for the environment. So our goal is to use basic catalyst that is environmental green.

Point 2: In the synthesis of pyridines and pyrazoles, the by-products should be referred in the manuscript.

Response 2: Based on TLC results for monitoring of the reaction progress, the rest of the starting material is left over unreacted at the end of the reaction. Thus, no other by-products could be isolated. 

Point 3: 15%wt should be changed into 15wt%.

Response 3: Thank you very much for your accurate comment, the mistake is corrected throughout the article.

Point 4: The products should be detected by NMR.

Response 4: The products were detected by the usual analytical tools if they are new and the previously published ones just detected by m.p., mass and elemental analyses and matched with that previously reported in the literature (References are cited).

Point 5: Please provide some structure information of the spent chitosan-La2O3 composite catalyst.

Response 5: The recovered nanocomposite was tested by IR and SEM and both of the two analyses showed that no noticeable changes were obtained after the completion of the reaction.

Round 2

Reviewer 2 Report

At present, the revision of the article is not enough to meet the requirements of publication. Please answer all the questions carefully and supplement the relevant experimental data.

Author Response

Although the additional experiments concerning the use of La2O3 nanoparticles are not easy to handle (experimental treatments difficulty such as its Recycling and contamination of product). Upon the reviewer and editor request, the results of the catalytic study of the La2O3 only, for comparison, were added in the two tables and we have previously added the results of chitosan alone. Now, I guess all the reviewer comments have been replied. Thanks and Accept my regards  

Round 3

Reviewer 2 Report

The authors modify the manuscript according to the comments and it can be accepted.